# Patterns of Changes in Oncological Care due to COVID-19: Results of a Survey of Oncological Nurses and Physicians from the Region of Hanover, Germany

**DOI:** 10.3390/healthcare10010015

**Published:** 2021-12-22

**Authors:** Johannes Beller, Jürgen Schäfers, Siegfried Geyer, Jörg Haier, Jelena Epping

**Affiliations:** 1Comprehensive Cancer Center, Hannover Medical School, 30625 Hannover, Germany; beller.johannes@mh-hannover.de (J.B.); schaefers.juergen@mh-hannover.de (J.S.); epping.jelena@mh-hannover.de (J.E.); 2Medical Sociology Unit, Hannover Medical School, 30625 Hannover, Germany; geyer.siegfried@mh-hannover.de

**Keywords:** oncology, cancer, COVID-19, profiles, clusters

## Abstract

Background: Healthcare staff is confronted with intensive decisional conflicts during the pandemic. Due to the specific burden of this moral distress in oncology, the investigation aimed at quantification of these conflicts and identification of risk factors that determine the extent and severity of these conflicts. We examined the heterogeneity of changes in oncology care due to COVID-19. Methods: We conducted a survey of oncological physicians and nurses in the region of Hanover, Germany in the second half of 2020. Overall, *N* = 200 respondents, 54% nurses, were included in the sample. Indicators of changes in oncology care were used to determine profiles of changes. To characterize these profiles, a diverse set of variables, including decision conflicts, uncertainty, age, gender, work experience, changes in communication with patients, psychological distress, work stress, process organization, and personnel resources, was obtained. Latent class analysis was conducted to determine these latent profiles. Results: We found that three distinct profiles best described the overall changes in oncology care due to COVID-19 in our sample, with each profile being associated with specific characteristics: (1) “Few Changes in Oncology Care” profile with 33% of participants belonging to this profile, (2) “Medium Changes in Oncology Care” profile with 43% of participants, and (3) “Severe Changes in Oncology Care” profile (24%). Participants from these profiles significantly differed regarding their age, work experience, occupational group, the prevalence of decision conflicts, decision uncertainty, quality of communication with patients, and quality of process organization. Conclusions: Distinct profiles of change in oncology care due to COVID-19 can be identified. Most participants reported small to medium changes, while some participants also reported severe changes. Profiles also differed regarding their associated characteristics. As such, specific consequences for better pandemic preparedness can be derived based on the current study. Future studies should investigate the patterns of changes in routine care due to COVID-19.

## 1. Background

In addition to the direct burden of disease caused by COVID-19, the COVID-19 pandemic probably also led to indirect consequences for the medical care of the population. Emergency care had to be prioritized at the cost of other conditions, such as cancer care. For example, some planned medical treatments were postponed, and existing treatment regimens need to be adapted to the care situations in a pandemic [1]. In agreement with this proposition, some studies have reported strong deteriorations of oncology care during COVID-19, whereas others provide more heterogeneous findings [2,3,4,5,6,7]. For example, Brugel and colleagues analyzed changes in oncology care during the first half of 2020 using routinely collected data in France and found strong impacts of COVID-19 on oncology care in some cases like screenings, but low impacts in other areas, such as chemotherapy use [2]. Similarly, another study reported results of a global survey in which self-reported changes in oncology care regarding oncology centers were obtained [4]. The authors again reported changes in oncology care due to COVID-19 in some cases, but also note that these changes appeared to be widely varying in magnitude between aspects of oncological care and across survey respondents with relation to their individual perspectives and resulting decisional uncertainty and decisional conflicts. In the current study, we further investigated this heterogeneity among changes to oncology care due to COVID-19. Due to the specific burden of healthcare staff in oncology during the pandemic, our investigation aimed at quantification of individual perspectives on decisional conflicts and identification of risk factors that determine the extent and severity of these conflicts.

One approach, that can capture the potential heterogeneity of changes in oncology care is latent class analysis (LCA), which has also been recommended in the literature [8,9]. LCA is a subset of classification methods, especially applicable for questionnaire data [10]. LCA finds groups of cases with similar characteristics in multivariate categorical data. It has been favorably used in health care studies [8,11,12]. However, to our knowledge, no studies have used this approach to study the heterogeneity of changes in oncology care in response to COVID-19. This will be the focus of the current study. We asked: Which change profiles of cancer care due to COVID-19 can be identified? Thereby, the current study will be useful in understanding the high degree of variability in care changes due to COVID-19. Consequently, it will also inform policies with which severe impacts of pandemics on routine care can be mitigated.

## 2. Methods

### 2.1. Sample

Cross-sectional data from the OnCoVID study (Oncology Management during the COVID-19 Pandemic—Ethical, Law and Health-Economical Implications) were used. Data were collected via a pen and paper survey in the second half of 2020 of nurses and physicians in the region of Hannover, Germany. Nurses and physicians were contacted via a regional network of cooperating clinics and out-patient centers who were provided with questionnaires. The inclusion criterion for healthcare professionals was immediate clinical interactions within oncological patient care. Responders could include those with higher as well as lower hierarchical status, across the whole age range. Further, no limitations according to oncological discipline were made, such that nurses and physicians from potentially all oncological disciplines and all socio-demographic backgrounds could respond. A response rate of 45% resulted. Thematically, the questionnaire focused on collecting information regarding changes in oncology care in Germany during COVID-19 from the individual perspective of nurses and physicians. Missing values (0–11% per variable) were imputed using the modern nonparametric missForest algorithm which was specially developed for mixed-type data, like in the current case, and thus compares favorably with other imputation algorithms [13]. 

### 2.2. Variables

Changes in oncology care due to COVID-19 were operationalized by asking participants to indicate the degree to which the COVID-19 pandemic has led to changes in oncology care regarding prevention, curative therapy, advanced cancers, palliative care, and psychosocial care. Changes regarding each of these aspects could be judged on a scale from “not at all” (1) to “Completely” (5). Additionally, decision conflicts were operationalized by asking whether participants experienced decision conflicts in their oncological decision-making. Participants could choose to answer with either “yes” (1) or “no” (0). Decision uncertainty was operationalized by inquiring about the degree of unsureness participants experienced in their oncology care decisions during COVID-19. Answers could be given on a scale from “not at all” (1) to “completely” (5). Changes in quality of communication with patients were operationalized via one item inquiring about the quality of perceived changes in communication with patients during COVID-19. Answers could be given on a 1 to 5 scale indicating the degree to which the processes changed during COVID-19 from “very negatively” (1) over no changes (3) to very positively (5). Psychological distress was operationalized by asking how participants felt during the first phase of COVID-19, with regards to depressiveness, anxiety, loneliness and stress. Participants could respond with one of four answer options ranging from “not at all/seldom" (1) to “most of the time” (4). A distress mean score was calculated by computing the mean of the four single items. The workload was operationalized by inquiring about the degree to which one’s workload changed during COVID-19 with answer options ranging from “much less workload” (1) to “much more workload” (5). Process Organization was operationalized via one item inquiring about the quality of perceived changes in general oncology care processes during COVID-19. Answers could be given on a 1 to 5 scale indicating the degree to which the processes changed during COVID-19 from “very negatively” (1) over no changes (3) to very positively (5). Similarly, Personnel Resources was operationalized via one item inquiring about the quality of perceived changes in personnel resources during COVID-19. Answers could be given on a 1 to 5 scale indicating the degree to which the personnel resources changed during COVID-19 from “very negatively” (1) over “no changes” (3) to “very positively” (5). Additional variables were years of professional experience (measured numerically in years), gender (male (0), female (1)), age, hierarchical status (no leadership function (0), leadership function (1)) and occupational group (physicians (0), nurses (1)).

### 2.3. Data Analysis

First, descriptive statistics of all variables are reported. Then latent class analysis (LCA) was conducted. Latent class analysis is a statistical technique based on maximum likelihood estimation that identifies groups of similar cases, as defined by specific combinations of observed variables [10]. In the case of changes in oncology care, groups of participants with similar perceptions of changes in oncology care will be identified. Via LCA it is then possible to assign group membership to each participant by estimating the probability of the participant belonging to each subgroup. Participants are assigned to class memberships based on modal probabilities. Then, the prevalence of profiles is calculated and demographic/risk factors are compared between profiles. Thus, in reporting the results of LCA, first information about the optimal number of classes/profiles is presented, which decides the number of substrata in the sample. After the number of classes/profiles is determined, information on how to characterize the classes/profiles is presented. This concerns the indicators by which the LCA was conducted, displayed by a plot of class-specific response profiles. Furthermore, additional indicators, such as socio-demographic information, are used to characterize classes. Research practices regarding sample size in LCA have been inconsistent [14]. In a recent review, the authors have identified LCA sample sizes in the published literature in the range of *N* = 131–16280 [15]. Regarding the methodological research, findings have been consistent in that a larger sample size has been found to produce more accurate LCA analysis. As to the optimal sample size, rules of thumb have traditionally suggested a sample size of around 200–300 as sufficient. However, the optimal sample size seems to also depend on model complexity, as simpler models with few classes and few indicators (such as in this study) have also been found to be accurate with a sample size as small as 30 [16,17]. Thus, although a larger sample size might be preferred, we do not expect severe problems due to a low sample size. All statistical analyses were performed with R.

## 3. Results

Survey participants were on average 43.12 years old (SD = 10.99; Range from 21 to 67). Additionally, 67% of participants reported being female (33% male), and 54% of participants were nurses and 46% physicians, respectively. They had an average work experience of about 18 years, and 41% of participants had a leadership role (Table 1). Furthermore, participants reflected, on average, slight to medium changes in oncological care due to COVID-19 regarding prevention (M = 3.04; SD = 1.22), curative therapy (M = 2.16; SD = 1.14), treatment of advanced cancers (M = 2.20; SD = 1.12), palliative care (M = 2.46; SD = 1.16), and psychosocial cancer care (M = 3.08; SD = 1.22). Further and more detailed sample characteristics and reported oncological changes are displayed in Table 1 and Table 2.

Then, latent class analysis was conducted. As shown in Figure 1, a three-class solution was found to be optimal, as judged by the Bayesian Information Criterion (BIC). Based on the plot of the class-specific response averages, as shown in Figure 2, these profiles can be described as: “Few Changes” profile (33% of participants; reporting lowest changes in each aspect); “Medium Changes” profile (43% of participants; reporting average changes regarding each other indicator); and “Severe Changes” profile (24% of participants; reporting high changes regarding each indicator).

These profiles also differed regarding their associated socio-demographic/risk behavior characteristics, as seen in Table 3. Significant differences between profiles were observed regarding age, work experience, and group membership, decision conflicts, decision uncertainty, communication with patients, and process organization. Participants in the “Severe Changes” profile differed from the other profiles in that they were more likely to have decision conflicts, experienced more decision uncertainty, were more likely to be nurses, reported stronger deterioration of communication with patients and of process organization.

## 4. Discussion

This investigation was conducted to quantify the individual perspectives of oncological healthcare staff related to their perception of changes in cancer care due to the pandemic and resulting conflicts in their professional decisions. We investigated the heterogeneity in reported changes to oncology care due to COVID-19 using latent class analysis. Three clusters of changes in oncological care during COVID-19 emerged, which can be described as (1) “Few Changes in Oncology Care”, (2) “Medium Changes in Oncology Care”, and (3) “Severe Changes in Oncology Care”. The “Medium changes” profile was most prevalent in this sample (43%), followed by the “Few changes” profile (33%), and the “Severe changes” profile (24%). Therefore, although the overall changes in oncology care can be described as slight to moderate in our sample, on average, there were substantial variations, with about a quarter of participants experiencing severe changes.

Thus, the current results support those of previous studies, which also generally identified medium changes, with large variabilities. Also in line with previous studies, we found that changes in screening and psychosocial care were generally those aspects of oncology that were affected the most due to COVID-19 [2]. Given this accumulating evidence that screenings and aspects of supportive care were most affected, future studies should investigate how these important aspects of oncology could be made more resilient to pandemics in the future [18,19,20].

Moreover, the current study suggests that the changes in oncology care profiles were associated with specific characteristics. For example, participants belonging to the “Severe changes” profile were significantly more likely than participants in the other profiles to work as oncological nurses, to report stronger deteriorations of communication with patients and higher impact on clinical process organization during COVID-19. As such, strategies to improve process quality and robustness as well as patient relationship and communication might be one important factor that mitigates severe negative changes to oncology care during a pandemic [21,22]. Doing so might also mitigate the potential for decisional conflicts and decisional uncertainty—which were much more likely to be experienced by participants in this “severe changes” profile—and might thus also prevent worse mental health for oncological personnel as well as worse treatment outcomes for patients [23,24,25,26]. In opposite, working experience was highest in the “Few Changes” profile, but decision uncertainty and decision conflicts were most prevalent in the “Severe Changes” profile. Perhaps, oncology care professionals with high work experience might have been especially successful in mitigating the loss of evidence and clinical routine caused by the COVID-19 pandemic [27]. From this perspective, retaining experienced personal becomes essential, for example by improving the work environment in the form of higher job autonomy, higher job control and better teamwork [28]. Oncological care professionals had to make difficult decisions during the COVID-19 pandemic. Given that these decisions are often required to be made despite an unusually large lack of empirical evidence, decision uncertainty and decision conflicts frequently resulted potentially leading to diminished quality of care and increased mental health problems in oncology care professionals. According to this phenomenon, more intensive interdisciplinary and interprofessional interactions can likely fill the evidence gap and reduce the resulting moral distress by providing shared experience (evidence grad 5). Thus, strong pandemic cancer care ad-hoc recommendations might be implemented to create more resilient organizational processes and reduce the mental health burden of oncology care professionals.

There are some limitations to the current study. First, the survey addressed only the perspective of oncological staff and did not obtain information on how oncological patients might have experienced cancer care during the COVID-19 pandemic. However, adequately considering the patient perspective is seen as of utmost importance to high quality cancer care, and as such future studies on this topic are needed [29]. As a second limitation, the current study could only use self-reports of health. As such, results might suffer from known self-report biases [30]. However, analyses targeting ethical and normative consequences of the pandemic preparedness clearly depend on this methodological approach. Our aim was to analyze individual perspectives of cancer care by healthcare professionals on changes in oncological care due to COVID-19. The trial design focused only on a cross-sectional design, and we could only analyze perceived changes in oncological care retrospectively. As one potential complication, the memory of changes in oncological care could be biased by the current care situation. Additionally, survey findings of the implications of the COVID-19 pandemic, in general, might prove to be highly time-specific. Therefore, our results can only be taken to represent oncology care during the first period of the COVID-19 pandemic. It is very likely that the subsequent course of the pandemic resulted in variations of professional reflection, but the reported finding likely represents a critical time period in the early phase of a pandemic and should be considered as part of pandemic preparedness. Future studies may also use other data sources, such as routinely collected admissions and procedures data which might be better able to encapsulate the time-dependent changes of the COVID-19 pandemic [31]. However, objective caseload in cancer care and subjective reflection of resulting decisional conflicts and uncertainty by healthcare professionals need to be differentiated. In a similar vein, besides technical problems, a low sample size might have resulted in a failure to uncover classes with low memberships [16]. Therefore, we cannot rule out that there are small classes with even more severely affected oncological care professionals. This possibility should be investigated by future studies. However, accepting these limitations the current study could further contribute to the understanding of the heterogeneous changes in oncology care due to COVID-19, which is seen as pivotal for future pandemic preparedness.

## Figures and Tables

**Figure 1 healthcare-10-00015-f001:**
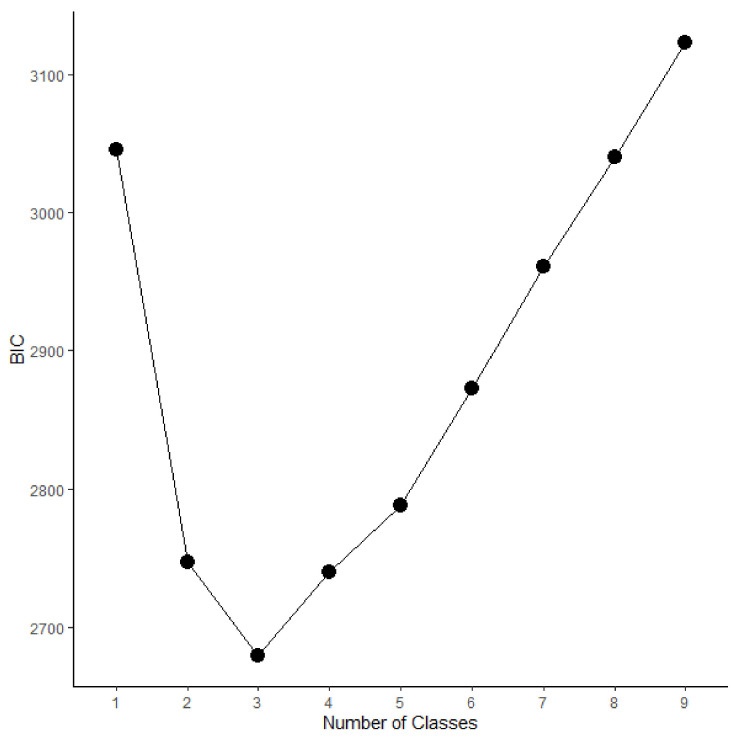
Scree-Plot of Latent Class Solutions with Different Number of Classes according to the BIC. Smaller values indicate a better fitting class-solution. Therefore, a three-class solution is found to fit best in this study.

**Figure 2 healthcare-10-00015-f002:**
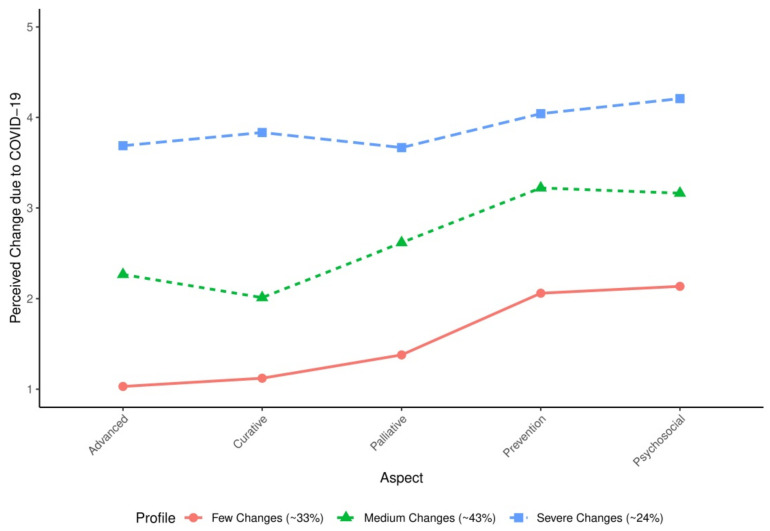
Latent Profiles with Class-Specific Response Probabilities and Population Proportions (*N* = 200). The Profile “Few Changes” includes about 33% of participants. Participants belonging to this profile are likely to report only small changes due to COVID-19, with reported change scores of about one to two in magnitude (on a scale of one to five). The Profile “Medium Changes” includes about 43% of participants (reported medium changes due to COVID-19 in their oncology care practice; change scores of about two to three in magnitude). The Profile “Severe Changes” includes about 24% of participants (reported severe changes due to COVID-19; change scores of about four in magnitude).

**Table 1 healthcare-10-00015-t001:** Sample Characteristics (*N* = 200).

Variable	Level	%/M (SD)
Age		43.16 (10.99)
Gender	Male	33.5%
	Female	66.5%
Work Experience		18.02 (11.66)
Leadership Role	No Leadership role	59.0%
	Leadership role	41.0%
Group	Physicians	46.5%
	Nurses	53.5%

**Table 2 healthcare-10-00015-t002:** Reported Change in Oncological Care (*N* = 200).

Variable	Level	%/M (SD)
Changes in Prevention	Not at all	15.0%
	To a small degree	18.0%
	To a medium degree	24.5%
	To a large degree	33.5%
	Completely	9.0%
Changes in Curative Therapy	Not at all	36.5%
	To a small degree	31.5%
	To a medium degree	14.0%
	To a large degree	16.0%
	Completely	2.0%
Changes in Advanced Cancer Care	Not at all	33.5%
	To a small degree	31.0%
	To a medium degree	20.5%
	To a large degree	12.0%
	Completely	3.0%
Changes in Palliative Care	Not at all	24.5%
	To a small degree	31.0%
	To a medium degree	23.0%
	To a large degree	17.0%
	Completely	4.5%
Changes in Psychosocial Care	Not at all	13.5%
	To a small degree	18.5%
	To a medium degree	27.0%
	To a large degree	29.0%
	Completely	12.0%
Decision Conflicts	Yes	67.0%
	No	33.0%
Decision Uncertainty	Not at all	23.5%
	To a small degree	41.0%
	To a medium degree	20.0%
	To a large degree	13.5%
	Completely	2.0%
Changes in Communication with Patients	Much worse	5.0%
	Slightly worse	27.5%
	About the same	57.0%
	Slightly better	9.0%
	Much better	1.5%
Psychological distress		1.78 (0.59)
Changes in Work stress	Much worse	4.0%
	Slightly worse	14.0%
	About the same	22.5%
	Slightly better	40.5%
	Much better	19.0%
Changes in Process Quality	Much worse	12.5%
	Slightly worse	47.0%
	About the same	26.5%
	Slightly better	13.0%
	Much better	1.0%
Changes in Personnel Resources	Much worse	9.5%
	Slightly worse	38.5%
	About the same	43.5%
	Slightly better	7.5%
	Much better	1.0%

**Table 3 healthcare-10-00015-t003:** Characteristics Across Profiles (*N* = 200).

Socio-Demographic/Risk Behavior Characteristics	Few Changes Profile	Medium Changes Profile	Severe Changes Profile	*p*
Age (M ± SD)	45.98 ± 10.8	40.6 ± 10.76	43.85 ± 10.80	0.007
Gender (%)	71%	62%	69%	0.431
Work Experience (M ± SD)	21.03 ± 12.34	14.92 ± 10.78	19.44 ± 11.06	0.006
Leadership Role (leadership %)	44%	41%	38%	0.786
Group (Nurses %)	59%	40%	71%	0.001
Decision Conflicts (Yes %)	18%	35%	50%	0.002
Decision Uncertainty (M ± SD)	1.85 ± 1.06	2.28 ± 0.85	2.94 ± 1.00	<0.001
Communication with Patients (M ± SD)	2.91 ± 0.57	2.7 ± 0.80	2.60 ± 0.84	0.013
Psychological Distress (M ± SD)	1.70 ± 0.52	1.80 ± 0.61	1.85 ± 0.62	0.554
Work Stress (M ± SD)	3.62 ± 1.05	3.53 ± 1.04	3.54 ± 1.18	0.916
Process Organization (M ± SD)	2.68 ± 0.9	2.31 ± 0.86	2.29 ± 0.94	0.012
Personnel Resources (M ± SD)	2.65 ± 0.71	2.47 ± 0.82	2.44 ± 0.90	0.138

Notes. Mean and Standard Deviations are displayed for ordinal variables in order to ease interpretation of group differences. *p* denotes *p*-values for comparisons between groups regarding the specific variable using either a χ^2^-Test in the case of binary variables, or a Kruskal-Wallis-Test in all other instances.

## Data Availability

The data that support the findings of this study are restricted due to privacy and ethical concerns.

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
