# Peer review of "Patterns of Changes in Oncological Care due to COVID-19: Results of a Survey of Oncological Nurses and Physicians from the Region of Hanover, Germany"

_healthcare, 2021, doi:10.3390/healthcare10010015_

Round 1

Reviewer 1 Report

The topic of the manuscript is very relevant and important not only for scientists, but also for practitioners. 

In the part Methods the variables of the research are described clearly but more information is needed about the sample (sociodemographic characteristics, are these nurses working in the similar or different oncology care departments, hierarchical status and etc.). Some of this information is in part Results, but it is not enough to understand the characteristics of the sample.

How to understand this sentence: Survey participants were on average 43.12 years old (SD = 10.99; Range from 21 to 67), with 67% being female, and 54% being nurses.  The sentences should be clear, explicit in the text.

In table 1 you are showing 2 different information - 1) there are sociodemographic characteristics and 2) evaluation of changes. For readers it would be easier to understand this information if you will do 2 separate tables.

Results which are shown in the figure 1 and 2 should be described in the text of the manuscript. Now the text about results is very inconsistent, fragmented. You should describe profiles, because from the figures reader cannot understand how these profiles differed regarding sociodemographic/risk behaviour characteristics. Also, please understand that the readers of your article will not necessarily be scientists versed/experienced in data analysis techniques. Kruskal-Wallis-Test let see the differences among 3 groups, but do not let to identify in what concrete groups these differences are statistically significant. You need to use post hoc test or other methods.

In summary, a very interesting study was done, valuable results were obtained, but the manuscript lacks completeness, depth, especially in the parts of the results and discussion. The conclusions of the study should also be formulated.

Author Response

Comment 1: The topic of the manuscript is very relevant and important not only for scientists, but also for practitioners.

Answer 1: We want to thank you for your constructive comments. Below you can find point-by-point responses to your remaining concerns and suggestions, in which we note changes made to the manuscript in line with your recommendations.

Comment 2: In the part Methods the variables of the research are described clearly but more information is needed about the sample (sociodemographic characteristics, are these nurses working in the similar or different oncology care departments, hierarchical status and etc.). Some of this information is in part Results, but it is not enough to understand the characteristics of the sample.

Answer 2: We extended our methods section with regards to the sample. The respective part now reads: “Cross-sectional data from the OnCoVID study (Oncology Management during the Covid-19 Pandemic - Ethical, Law and Health-Economical Implications) were used. Data were collected via a pen and paper survey in the second half of 2020 of nurses and physicians in the region of Hannover, Germany. Nurses and physicians were contacted via a regional network of cooperating clinics and out-patient centers who were provided with questionnaires. Inclusion criterion for healthcare professionals was immediate clinical interactions within oncological patient care. Responders could include those with higher as well as lower hierarchical status, across the whole age range. Further, no limitations according to oncological discipline were made, such that nurses and physicians from potentially all oncological disciplines and all socio-demographic backgrounds could respond. A response rate of 45% resulted. Thematically, the questionnaire focused on collecting information regarding changes in oncology care in Germany during COVID-19 from the individual perspective of nurses and physicians. Missing values (0%–11% per variable) were imputed using the modern nonparametric missForest algorithm which was especially developed for mixed-type data, like in the current case, and thus compares favorably with other imputation algorithms [13].”.

Comment 3: How to understand this sentence: Survey participants were on average 43.12 years old (SD = 10.99; Range from 21 to 67), with 67% being female, and 54% being nurses.  The sentences should be clear, explicit in the text.

Answer 3: We revised the sentence you refer to. It now reads: “Survey participants were on average 43.12 years old (SD = 10.99; Range from 21 to 67). Additionally, 67% of participants reported being female (33% male), and 54% of participants were nurses and 46% physicians, respectively. They had an average work experience of about 18 years, and 41% of participants had a leadership role (Table 1).”.

Comment 4: In table 1 you are showing 2 different information - 1) there are sociodemographic characteristics and 2) evaluation of changes. For readers it would be easier to understand this information if you will do 2 separate tables.

Answer 4: In accordance with your comment, we now provide two different tables, one with sociodemographic information (Table 1), the other with the evaluation of changes (Table 2).

Comment 5: Results which are shown in the figure 1 and 2 should be described in the text of the manuscript. Now the text about results is very inconsistent, fragmented. You should describe profiles, because from the figures reader cannot understand how these profiles differed regarding sociodemographic/risk behaviour characteristics. Also, please understand that the readers of your article will not necessarily be scientists versed/experienced in data analysis techniques. Kruskal-Wallis-Test let see the differences among 3 groups, but do not let to identify in what concrete groups these differences are statistically significant. You need to use post hoc test or other methods.

Answer 5: We now explicitly describe the results shown in Figure 1 and 2: “Figure 1. Scree-Plot of Latent Class Solutions with Different Number of Classes according to the BIC. Smaller values indicate a better fitting class-solution. Therefore, a three-class solution is found to fit best in this study.“. Furthermore: “Figure 2. Latent Profiles with Class-Specific Response Probabilities and Population Proportions (N = 200). The Profile “Few Changes” includes about 33% of participants. Participants belonging to this profile are likely to report only small changes due to COVID-19, with reported change scores of about one to two in magnitude (on a scale of one to five). The Profile “Medium Changes” includes about 43% of participants (reported medium changes due to COVID-19 in their oncology care practice; change scores of about two to three in magnitude). The Profile “Severe Changes” includes about 24% of participants (reported severe changes due to COVID-19; change scores of about four in magnitude).”. Regarding the post-hoc tests we would prefer not to conduct post-hoc tests as we think that this would constitute too much statistical tests (and inflate alpha) and might thus make the manuscript harder to understand. Additionally, as you suggested we now provide more detail as to the data-analysis in our study: “First, descriptive statistics of all variables are reported. Then latent class analysis (LCA) was conducted. Latent class analysis is a statistical technique based on maximum likelihood estimation that identifies groups of similar cases, as defined by specific combinations of observed variables [10]. In the case of changes in oncology care, groups of participants with similar perceptions of changes in oncology care will be identified. Via LCA it is then possible to assign group membership to each participant by estimating the probability of the participant belonging to each subgroup. Participants are assigned to class memberships based on modal probabilities. Then, prevalence of profiles are calculated and demographic/risk factors are compared between profiles. Thus, in reporting the results of LCA, first information about the optimal number of classes/profiles is presented, which decides about the number of substrata in the sample. After the number of classes/profiles is determined, information on how to characterize the classes/profiles is presented. This concerns the indicators by which the LCA was conducted, displayed by a plot of class-specific response profiles. Furthermore, additional indicators, such as socio-demographic information, is used to characterize classes. Research practices regarding sample size in LCA have been inconsistent [14]. In a recent review, the authors have identified LCA sample sizes in the published literature in the range of N = 131-16280 [15]. Regarding the methodological research, findings have been consistent in that a larger sample size has been found to produce more accurate LCA analysis. As to the optimal sample size, rules of thumb have traditionally suggested a sample size of around 200-300 as sufficient. However, the optimal sample size seems to also depend on model complexity, as simpler models with few classes and few indicators (such as in this study) have also been found to be accurate with a sample size as small as 30 [16, 17]. Thus, although a larger sample size might be preferred, we do not expect severe problems due to a low sample size. All statistical analyses were performed with R.”.

Comment 6: In summary, a very interesting study was done, valuable results were obtained, but the manuscript lacks completeness, depth, especially in the parts of the results and discussion. The conclusions of the study should also be formulated.

Answer 6: Thank you again for commenting on our manuscript. We have revised the manuscript in accordance with your comments and hope that the revisions in the manuscript along with our answers to your comments can alleviate your concerns.

Reviewer 2 Report

given the underpowered sample of study and the lack of validity and reliability for the measurement it is hard to know if the findings are believable.

Author Response

Comment 1: Given the underpowered sample of study and the lack of validity and reliability for the measurement it is hard to know if the findings are believable

Answer 1: We want to thank you for your comment. We have added methodological information regarding the power of our analysis to the methods section, emphasizing why we believe that the results are valid and reliable: “Research practices regarding sample size in LCA have been inconsistent [14]. In a recent review, the authors have identified LCA sample sizes in the published literature in the range of N = 131-16280 [15]. Regarding the methodological research, findings have been consistent in that a larger sample size has been found to produce more accurate LCA analysis. As to the optimal sample size, rules of thumb have traditionally suggested a sample size of around 200-300 as sufficient. However, the optimal sample size seems to also depend on model complexity, as simpler models with few classes and few indicators (such as in this study) have also been found to be accurate with a sample size as small as 30 [16, 17]. Thus, although a larger sample size might be preferred, we do not expect severe problems due to a low sample size.”. Additionally, we now emphasize this potential limitation in the discussion: “In a similar vein, besides technical problems, a low sample size might have resulted in a failure to uncover classes with low memberships [16]. Therefore, we cannot rule out that there are small classes with even more severely affected oncological care professionals. This possibility should be investigated by future studies.”.

Thank you again for your comment.

Reviewer 3 Report

This paper tackles an important topic related to healthcare during the covid19 pandemic. The paper is well written and the results can be of interest for practice. Please find below my remarks to further improve the manuscript.

  1. Data are cross-sectional. No longitudinal (e.g., pre-post) design has been used. Participants were asked to retrospectively self-report the respective changes. These limitations need to be discussed in more detail.

  1. Likewise, given that no true changes could be investigated across multiple measurement points, the potential confound of short-term changes with fluctuations over time need to be addressed.

  1. The latent analysis is nice, containing several important predictors. However, is the sample of 200 participants large enough to reliably conduct these complex analyses?

  1. The practical implications could be illustrated with more detailed examples from real working life. For example, which personal, work-place-related, and organizational changes need to be implemented to allow care workers to face their particular job demands and risks for suffering from burnout, depression, etc. as well as to assure the maintenance of high-standard healthcare services during the specific situation of a pandemic?

Author Response

Comment 1: This paper tackles an important topic related to healthcare during the covid19 pandemic. The paper is well written and the results can be of interest for practice. Please find below my remarks to further improve the manuscript.

Answer 1: We want to thank you for your constructive comments. Below you can find point-by-point responses to your remaining concerns and suggestions, in which we note changes made to the manuscript in line with your recommendations.

Comment 2: Data are cross-sectional. No longitudinal (e.g., pre-post) design has been used. Participants were asked to retrospectively self-report the respective changes. These limitations need to be discussed in more detail. Likewise, given that no true changes could be investigated across multiple measurement points, the potential confound of short-term changes with fluctuations over time need to be addressed.

Answer 2: We now discuss these points in more detail in the limitations section: “Our aim was to analyze individual perspectives of cancer care by healthcare professionals on changes in oncological care due to COVID-19. The trial design focused only on cross-sectional such that we could only analyze perceived changes in oncological care retrospectively. As one potential complication, memory of changes in oncological care could be biased by the current care situation. Additionally, survey findings about the implications of the COVID-19 pandemic, in general, might prove to be highly time-specific. As such our results can only be taken to provide one snapshot about oncology care during the first period of the COVID-19 pandemic. It is very likely that the subsequent course of the pandemic resulted in variations of professional reflection, but the reported finding likely represent a critical time period in the early phase of a pandemic and should be considered as part of pandemic preparedness. Future studies may also use other data sources, such as routinely collected admissions and procedures data which might be better able to encapsulate the time-dependent changes of the COVID-19 pandemic [e.g., 31]. However, objective case load in cancer care and subjective reflection of resulting decisional conflicts and uncertainty by healthcare professionals need to be differentiated.”.

Comment 3: The latent analysis is nice, containing several important predictors. However, is the sample of 200 participants large enough to reliably conduct these complex analyses?

Answer 3: In accordance with reviewer 2 we have expanded our introduction to the latent class analysis, especially regarding sample size: “Research practices regarding sample size in LCA have been inconsistent [14]. In a recent review, the authors have identified LCA sample sizes in the published literature in the range of N = 131-16280 [15]. Regarding the methodological research, findings have been consistent in that a larger sample size has been found to produce more accurate LCA analysis. As to the optimal sample size, rules of thumb have traditionally suggested a sample size of around 200-300 as sufficient. However, the optimal sample size seems to also depend on model complexity, as simpler models with few classes and few indicators (such as in this study) have also been found to be accurate with a sample size as small as 30 [16, 17]. Thus, although a larger sample size might be preferred, we do not expect severe problems due to a low sample size.”. Furthermore, we now point to the need to replicate our analyses with a larger sample size: “In a similar vein, besides technical problems, a low sample size might have resulted in a failure to uncover classes with low memberships [16]. Therefore, we cannot rule out that there are small classes with even more severely affected oncological care professionals. This possibility should be investigated by future studies.”.

Comment 4: The practical implications could be illustrated with more detailed examples from real working life. For example, which personal, work-place-related, and organizational changes need to be implemented to allow care workers to face their particular job demands and risks for suffering from burnout, depression, etc. as well as to assure the maintenance of high-standard healthcare services during the specific situation of a pandemic?

Answer 4: We have expanded our discussion of practical implications, in accordance with your comments: “Moreover, the current study suggests that the changes in oncology care profiles are associated with specific characteristics. For example, participants belonging to the “Severe changes” profile were significantly more likely than participants in the other profiles to work as oncological nurses, to report stronger deteriorations of communication with patients and higher impact on clinical process organization during COVID-19. As such, strategies to improve process quality and robustness as well as patient relationship and communication might be one important factor that mitigate severe negative changes to oncology care during a pandemic [21, 22]. Doing so might also mitigate the potential for decisional conflicts and decisional uncertainty—which were much more likely to be experienced by participants in this “severe changes” profile—and might thus also prevent worse mental health for oncological personnel as well as worse treatment outcomes for patients [23–26]. In opposite, working experience was highest in the “Few Changes” profile, but decision uncertainty and decision conflicts were most prevalent in the “Severe Changes” profile. Perhaps, oncology care professionals with high work experience might have been especially successful in mitigating the loss of evidence and clinical routine caused by the COVID-19 pandemic [27]. From this perspective, retaining experienced personal becomes essential, for example by improving the work environment in the form of higher job-autonomy, higher job-control and better teamwork [28]. Oncological care professionals had to make difficult decisions during the COVID-19 pandemic. Given that these decisions often need to be made despite an unusual large lack of empirical evidence, decision uncertainty and decision conflicts frequently resulted potentially leading to diminished quality of care and increased mental health problems in oncology care professionals. According to this phenomenon, more intensive interdisciplinary and interprofessional interactions can likely fill the evidence gap and reduce the resulting moral distress by providing shared experience (evidence grad 5).  Thus, strong pandemic cancer care ad-hoc recommendations might be implemented to create more resilient organizational processes and reduce mental health burden of oncology care professionals.”.

Thank you again for reviewing our manuscript.

Round 2

Reviewer 1 Report

The authors made corrections of the manuscript. Now I suppose, the manuscript could be published.

Author Response

Thank you for this suggestion. We have included dedicated sentences about the aim of our investigation within the abstract, introduction and discussion.